# COVID-19 in Portugal: a retrospective review of paediatric cases, hospital and PICU admissions in the first pandemic year

Cecilia Elias [1,2] Rodrigo Feteira-Santos [1,3,4,5] Catarina Camarinha,[1] Miguel de Araújo Nobre [1,6] Andreia Silva Costa [1,4,5,7,8] Leonor Bacelar-Nicolau [1,3,4,9] Cristina Furtado [1,4,5,9,10] Paulo Jorge Nogueira [1,3,4,5,7,9,11]

## ABSTRACT

**Background** COVID-19 is considered by WHO a pandemic with public health emergency repercussions. Children often develop a mild disease with good prognosis and the recognition of children at risk is essential to successfully manage paediatric COVID-19. Quality epidemiological surveillance data are required to characterise and assess the pandemic.

**Methods** Data on all reported paediatric COVID-19 cases, in Portugal, were retrospectively assessed from a fully anonymised dataset provided by the Directorate General for Health (DGS). Paediatric hospital admission results were obtained from the DGS vaccine recommendations and paediatric intensive care unit (PICU) admission results from the EPICENTRE.PT group. Reported cases and PICU admissions from March 2020 to February 2021 and hospital admissions between March and December 2020 were analysed.

**Results** 92 051 COVID-19 cases were studied, 50.5% males, average age of 10.1 years, corresponding to 5.4% of children in Portugal. The most common symptoms were cough and fever, whereas gastrointestinal symptoms were infrequent. The most common comorbidity was asthma. A high rate of missing surveillance data was noticed, on presentation of disease and comorbidity variables, which warrants a cautious interpretation of results. Hospital admission was required in 0.93% of cases and PICU on 3.48 per 10 000 cases. PICU admission for Multisystem Inflammatory Syndrome in Children (MIS-C) was more frequent in children with no comorbidities and males, severe COVID-19 was rarer and occurred mainly in females and infants. Case fatality rate and mortality rates were low, 1.8 per 100 000 cases and 1.2 per 1 000 000 cases, respectively.

**Conclusions** The overall reported case incidence was 5.4 per 100 children and adolescents and <1% of cases required hospital admission. MIS-C was more frequent in patients with no comorbidities and males. Mortality and case fatality rates were low. Geographic adapted strategies, and information systems to facilitate surveillance are required to improve surveillance data quality.

For numbered affiliations see end of article.

**Correspondence to**
Dr Cecilia Elias; cecilia.elias@proton.me

### WHAT IS ALREADY KNOWN ON THIS TOPIC
⇒ Quality health surveillance data are essential to assess the COVID-19 pandemic.
⇒ Children present a milder course of disease and a lower susceptibility to being severely affected by COVID-19.
⇒ Globally, paediatric COVID-19 case fatality and mortality rates are low.

### WHAT THIS STUDY ADDS
⇒ Portuguese paediatric surveillance data and systems need improvement.
⇒ Paediatric COVID-19 incidence was 5.4% during the first pandemic year.
⇒ Portuguese children and adolescents presented an extremely low mortality and case fatality rate, 1.8 per 100 000 cases and 1.2 per 1 000 000 cases, respectively.

### HOW THIS STUDY MIGHT AFFECT RESEARCH, PRACTICE AND/OR POLICY
⇒ National paediatric COVID-19 results allow for improvement at a country level.
⇒ Future research should focus on health surveillance improvement.

## BACKGROUND

COVID-19, caused by SARS-CoV-2, is considered by WHO a pandemic with public health emergency repercussions.[1 2] Worldwide, millions of people have been infected with COVID-19 since the end of 2019 among all age groups.[3] Children are also at risk for COVID-19 but have experienced a lower incidence than other age groups.[4 5] Laboratory-confirmed COVID-19 cases in the USA reveal that only 1.7% of cases were among children and adolescents under 18 years of age.[6] Furthermore, several studies showed these age groups have lower susceptibility to being severely affected, often developing a

mild form of disease and with a likely good prognosis.[7–9] Studies have shown that children present different symptoms, distinct comorbidities, decreased disease severity and fewer disease complications.[10] The characteristics and clinical manifestations of infected children[11 12] have been considered of major interest.[13] The most frequent symptoms described in children have been fever, cough[14] and ear, nose and throat complaints. However, other manifestations such as gastrointestinal symptoms have also been identified.[12 15 16] A small number of children needed hospital admission, and of these, a minority required intensive care unit (ICU) admission either due to severe COVID-19 or multisystem inflammatory disease in children (MIS-C).[17 18] Several risk factors have been associated with these conditions such as extreme age, male sex, comorbidities and non-white background.[19]

As COVID-19 characteristics and clinical course among children can differ from adult patients in numerous ways, specific recommendations for the diagnosis, treatment and vaccination plans in children are being implemented worldwide, despite differences across countries.[20–23] Increasing knowledge of the clinical characteristics in this demographic group may also contribute towards the development and continuous adaptation of preventive strategies that can benefit all communities. Epidemiological surveillance data are essential to attain these objectives. Epidemiological surveillance is a critical part of public health practice,[24] where continuous collection, analysis and interpretation of health data help identify disease clusters, assess health trends, monitor interventions, identify high-risk groups and guide decision making and action.[25] Furthermore, epidemiological data quality is essential and should be monitored to guarantee data results are meaningful.[26]

This study aimed to investigate the epidemiological and clinical characteristics of paediatric SARS-CoV-2 infection cases, hospital and ICU admissions, compare these with other countries and to assess the quality of epidemiological surveillance data in Portugal.

## MATERIALS AND METHODS
### Study design and data
Data on all cases of COVID-19 in children and adolescents under the age of 18 years, in Portugal, diagnosed between 2 March 2020 and 28 February 2021 were retrospectively assessed. The dataset was fully anonymised and was provided by the Directorate General for Health (DGS) on 11 March 2021. The data were collected by the SINAVE (National Epidemiological Surveillance System). The SINAVE is an online system aimed at the electronic notification of mandatory reporting diseases in Portugal. The SINAVE database includes demographic and clinical information collected from all reported patients. Epidemiological surveillance data in SINAVE are meant to be filled by public health doctors and nurses. However, due to the COVID-19 pandemic increased workload, other professionals from a diverse background have also been recruited to support in surveillance. According to the WHO definition, a confirmed SARS-CoV-2 case was based on a positive PCR test.[27] Hospital admissions with a SARS-CoV-2 primary or secondary diagnosis, from 2 March to 31 December 2020, were obtained as results from the paediatric COVID-19 vaccination technical recommendation by DGS.[28] These results include admissions to all hospitals in Portugal with a SARS-CoV-2 diagnosis. No information on whether the SARS-CoV-2 diagnosis was the primary or secondary cause of admission was available. Data on paediatric SARS-CoV-2 paediatric intensive care unit (PICU) admissions, from March 2020 to February 2021, were obtained as anonymised data results on severe COVID-19[29] and MIS-C[30] from the EPICENTRE. PT work group. Daily death reports published by DGS were analysed to calculate the number of deaths caused by COVID-19 and these overlapped with the recorded SINAVE case deaths. Data from the National Statistics Institute were used for mortality and population comparisons. The paediatric mid-2020 population was used by proxy as an estimate population for the period analysed.

### Patient and public involvement
Patients and/or the public were not involved in the design, or conduct, or reporting, or dissemination plans of this research.

### Variables
The SINAVE system includes two variables which indicate the patient's clinical presentation (as symptomatic or asymptomatic) and comorbidities (present, absent). Information on symptoms and comorbidities associated with each case notification should also be filled in through individual variables. For a given symptom or comorbidity, health professionals selected the respective field as yes or no, according to the patient's information. However, any information not filled, for a given symptom or comorbidity, created a missing value in the database. Considering this, a new variable was calculated to describe if a patient was symptomatic or not, using the patient's clinical presentation variable and associated symptoms variables and similarly for the presence of comorbidities. A patient was considered symptomatic either if health professionals classified it as symptomatic through the SINAVE original variable or at least one symptom was reported. In addition, those with any symptom variable filled in but not previously recorded as symptomatic were considered symptomatic patients. Patients' symptomatology status was missing when no symptom variables were filled, and no information regarding the presentation of symptoms in the original variable. A patient was considered having comorbidities either if the presence of comorbidities was selected through the SINAVE original variable or at least one comorbidity was reported.

### Statistical analysis
Descriptive statistics were used to summarise univariate characteristics. Results were presented as absolute

**Table 1**  Age and sex of paediatric COVID-19-infected cases, from March 2020 to February 2021 in Portugal

|  |  | Total n (%)* | Female n (%)† | Male n (%)† | Reported cases per 100 children (%) |
|---|---|---|---|---|---|
| N (%) |  | 92 051 | 45 583 | 46 453 | 5.4 |
|  | 0 | 1719 (1.9%) | 826 (1.8%) | 884 (1.9%) | 2.0 |
|  | 1 | 3725 (4.0%) | 1784 (3.9%) | 1940 (4.2%) | 4.3 |
|  | 2 | 3692 (4.0%) | 1860 (4.1%) | 1832 (3.9%) | 4.2 |
|  | 3 | 3332 (3.6%) | 1575 (3.5%) | 1757 (3.8%) | 3.8 |
|  | 4 | 3709 (4.0%) | 1840 (4.0%) | 1869 (4.0%) | 4.2 |
|  | Subtotal 1–4 years | 14 458 (15.7%) | 7059 (15.5%) | 7398 (15.9%) | 4.1 |
|  | 5 | 4023 (4.4%) | 1988 (4.4%) | 2035 (4.4%) | 4.6 |
|  | 6 | 4299 (4.7%) | 2128 (4.7%) | 2171 (4.7%) | 5.1 |
|  | 7 | 4592 (5.0%) | 2232 (4.9%) | 2360 (5.1%) | 5.4 |
|  | 8 | 5058 (5.5%) | 2490 (5.5%) | 2568 (5.5%) | 5.5 |
| Age | 9 | 5548 (6.0%) | 2709 (5.9%) | 2839 (6.1%) | 5.7 |
|  | Subtotal 5–9 years | 23 520 (25.6%) | 11 547 (25.3%) | 11 973 (25.8%) | 5.3 |
|  | 10 | 5705 (6.2%) | 2872 (6.3%) | 2833 (6.1%) | 5.6 |
|  | 11 | 5726 (6.2%) | 2838 (6.2%) | 2888 (6.2%) | 5.9 |
|  | 12 | 5865 (6.4%) | 2928 (6.4%) | 2937 (6.3%) | 5.8 |
|  | 13 | 6080 (6.6%) | 2942 (6.5%) | 3137 (6.8%) | 6.1 |
|  | 14 | 6449 (7.0%) | 3235 (7.1%) | 3213 (6.9%) | 6.3 |
|  | Subtotal 10–14 years | 29 825 (32.4%) | 14 815 (32.5%) | 15 008 (32.3%) | 5.9 |
|  | 15 | 6919 (7.5%) | 3402 (7.5%) | 3514 (7.6%) | 6.5 |
|  | 16 | 7635 (8.3%) | 3890 (8.5%) | 3744 (8.1%) | 7.3 |
|  | 17 | 7975 (8.7%) | 4041 (8.9%) | 3932 (8.5%) | 7.4 |
|  | Subtotal 15–17 years | 22 529 (24.5%) | 11 333 (24.9%) | 11 190 (24.1%) | 7.1 |

Fifteen (0.02%) missing values were observed for sex.
*Global percentage.
†Percentage within sex level.

and relative frequencies for qualitative variables, and as means, medians and SD for quantitative variables. Calculated percentages have excluded missing cases unless otherwise stated. All analysis were conducted using R software V.3.6.3 (R Foundation for Statistical Computing, Vienna, Austria) and IBM SPSS Statistics V.28.0.1.1 (14).

## RESULTS
### General characteristics
The present study included 92 051 COVID-19 cases in children and adolescents reported from 2 March 2020 to 28 February 2021, in Portugal; 46 453 (50.5%) were males and 45 583 (49.5%) females, the overall reported case incidence was 5.4 per 100 children and adolescents and corresponding to 5.3% of males and 5.5% of females. The age profile of patients was distributed between 0 and 17 years old, with the number of cases increasing with age (table 1). An average age of 10.1 (SD 5.0) and a median of 11.0 years were observed.

The geographical distribution highlights most cases occurred in three urban areas, the Lisbon district (n=22 999, 25%), followed by Porto (n=18 350, 19.9%) and Braga (n=11 091, 12%). The areas with fewer cases were predominantly rural districts (Portalegre and Beja) and autonomous regions of the Azores and Madeira (figure 1). The children and adolescents with COVID-19 had mostly Portuguese nationality (n=88 387, 95.3%).

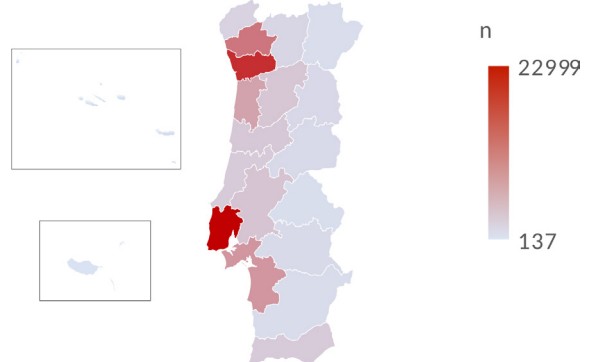

**Figure 1**  Map of COVID-19 paediatric cases in Portugal, by district in absolute value. (top inset box - Azores, bottom inset box - Madeira)

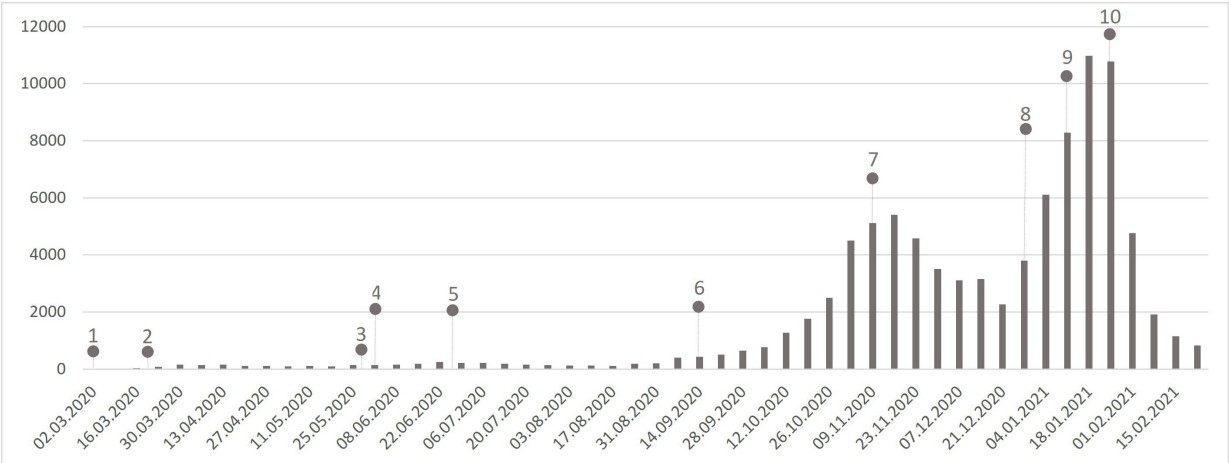

**Figure 2** Paediatric COVID-19 cases per week. 1. First patient with COVID-19 diagnosed in Portugal; 2. First national state of emergency; 3. Lockdown easing; 4. Preschool and primary school reopening; 5. End of the school year; 6. Start of the school year; 7. State of emergency with restrictions; 8. Christmas with minimal gathering restrictions; 9. Reintroduction of mandatory remote working, travel restrictions and school closures; 10. Highest daily death toll due to COVID-19 during the pandemic.

Only 4% were from other backgrounds, mainly from Brazil, Angola, Cape Verde, São Tome and Principe and scattered from a wide range of countries.

### COVID-19 cases timeline

During this study, several COVID-19 epidemic waves were observed (figure 2): (1) between March and April 2020, when 738 cases were reported (0.8% of all cases); (2) between May and July 2020, when 1756 cases (1.9% of all cases) were reported; (3) from September to December 2020, when 39 897 cases (43.3% of all cases) were reported and (4) from January to March 2021, when 46 634 cases (50.7% of all cases) were reported.

### Clinical presentation

Information on disease presentation, as symptomatic or asymptomatic, was reported in 47 470 (52%) cases while there was no information regarding the presence or absence of symptoms in 44 192 cases (48%). From cases with information on clinical presentation, 13 077 (27.5%) were asymptomatic and 34 393 (72.5%) symptomatic. Specific symptoms were reported in 21 077 (61.3%) cases and a further 13 316 (38.7%) cases were recorded as symptomatic without reporting symptoms. The asymptomatic presentation of the disease increased with age, peaking between 4 and 10 years of age and slightly decreasing subsequently.

To describe the symptomatic presentation, an analysis which included only cases where symptoms were recorded was performed (n=21 077). The most common symptoms reported were cough (n=10 205, 48.4%), fever (n=9812, 46.6%), rhinorrhoea (n=7739, 36.7%) and headache (n=7143, 33.9%). Gastrointestinal symptoms, such as abdominal pain, nausea and vomiting and diarrhoea were reported in 7.1%, 7.8% and 10.2%, respectively. Analysing the distribution of the co-occurrence of clinical COVID-19 symptoms, in each paediatric patient, fever was commonly combined with cough (38.2%),

rhinorrhoea (29.8%), headache (29%), odynophagia (21%) or myalgia (16.6%). Furthermore, combinations of these symptoms were the most common presentations reported. Fever, cough and rhinorrhoea were the most common presentations for children from 0 to 9 (table 2). Children aged zero presented the highest rates of fever (48%), cough (37%), rhinorrhoea (35%) and diarrhoea (24%) when compared with the other age groups. From age 10 to 17 years, headache was the most common reported symptom, followed by fever, cough and rhinorrhoea.

### Comorbidities

Information on the presence or absence of comorbidities was reported in 19 680 (21.4%) cases and there was no information regarding the presence or absence of comorbidities in 72 371 (78.6%). A total of 1593 patients presented with comorbidities, of which 1000 were specified. The three most frequently reported comorbidities in children and adolescents with COVID-19 were: asthma (n=753, 0.8%), neuromuscular diseases (n=49) and diabetes (n=55). Most patients with asthma or neurological disease were males, whereas most patients with diabetes were females (table 3).

### Hospital and PICU admissions and deaths

A total of 476 children were admitted to hospital with a COVID-19 International Classification of Diseases 10th Revision code, either as primary or secondary diagnosis. Children aged from 0 to 4 years represented 47.9% (228) of hospital admissions, corresponding to 2.24% of all COVID-19 cases in this age group during this time. Overall, 2.8 per 10 000 children in Portugal required hospital admission (table 4). No other sociodemographic or clinical variables were available to further characterise this population.

A total of 32 children required PICU admission, corresponding to 0.19 PICU admissions per 10 000 children

**Table 2** Symptoms associated with COVID-19 per age group from March 2020 to February 2021 in Portugal

| | Age groups (years) | | | | | |
| --- | --- | --- | --- | --- | --- | --- |
| | 0 | 1–4 | 5–9 | 10–14 | 15–17 | Total |
| | N (%) | N (%) | N (%) | N (%) | N (%) | N (%) |
| Total | 1075 | 8316 | 11 733 | 14 759 | 11 587 | 47 470 |
| Symptoms | | | | | | |
| Fever | 513 (47.7%) | 2846 (34.2%) | 2096 (17.9%) | 2317 (15.7%) | 1965 (17.0%) | 9737 (20.5%) |
| Cough | 396 (36.8%) | 2397 (28.8%) | 1920 (16.4%) | 2632 (17.8%) | 2771 (23.9%) | 10 116 (21.3%) |
| Dyspnoea | 34 (3.2%) | 192 (2.3%) | 136 (1.2%) | 180 (1.2%) | 258 (2.2%) | 800 (1.7%) |
| Rhinorrhoea | 376 (35.0%) | 2047 (24.6%) | 1465 (12.5%) | 1927 (13.1%) | 1818 (15.7%) | 7633 (16.1%) |
| Odynophagia | 13 (1.2%) | 364 (4.4%) | 1003 (8.5%) | 1747 (11.8%) | 1831 (15.8%) | 4958 (10.4%) |
| Headache | 1 (0.1%) | 149 (1.8%) | 1351 (11.5%) | 2723 (18.4%) | 2841 (24.5%) | 7065 (14.9%) |
| Abdominal pain | 8 (0.7%) | 199 (2.4%) | 553 (4.7%) | 449 (3.0%) | 252 (2.2%) | 1461 (3.1%) |
| Chest pain | 0 (0.0%) | 11 (0.1%) | 59 (0.5%) | 148 (1.0%) | 214 (1.8%) | 432 (0.9%) |
| Arthralgia | 0 (0.0%) | 9 (0.1%) | 26 (0.2%) | 64 (0.4%) | 61 (0.5%) | 160 (0.3%) |
| Myalgia | 1 (0.1%) | 84 (1.0%) | 436 (3.7%) | 1300 (8.8%) | 1731 (14.9%) | 3552 (7.5%) |
| Nausea and vomiting | 38 (3.5%) | 362 (4.4%) | 403 (3.4%) | 475 (3.2%) | 344 (3.0%) | 1622 (3.4%) |
| Diarrhoea | 107 (10.0%) | 616 (7.4%) | 443 (3.8%) | 523 (3.5%) | 464 (4.0%) | 2153 (4.5%) |
| Convulsions | 1 (0.1%) | 17 (0.2%) | 5 (0.0%) | 10 (0.1%) | 5(0.0%) | 38 (0.1%) |
| Irritability | 19 (1.8%) | 43 (0.5%) | 9 (0.1%) | 9 (0.1%) | 5 (0.0%) | 85 (0.2%) |
| General weakness | 16 (1.5%) | 148 (1.8%) | 221 (1.9%) | 533 (3.6%) | 724 (6.2%) | 1642 (3.5%) |
| Tachycardia | 2 (0.2%) | 20 (0.2%) | 15 (0.1%) | 16 (0.1%) | 12 (0.1%) | 65 (0.1%) |

Percentages of patients presenting a specific symptom from patients presenting with the column symptom.

and adolescents in Portugal. Of these, 28 (87.5%) were admitted due to MIS-C and 4 (12.5%) due to severe COVID-19. MIS-C patients were predominantly male (21, 75%) with ages from 15 months to 17 years of age, average of 10.75 and median of 10.5 years. Most cases occurred between January and February 2021. Six patients presented comorbidities. Patients with severe COVID-19 were mainly females, under the age of 1 and all presenting with comorbidities. Two deaths caused by COVID-19 were reported by the DGS (the National Health Authority) during this study and coincided with two deaths recorded in the SINAVE database.

## DISCUSSION

In this study, epidemiological and clinical characteristics of paediatric COVID-19 infections, hospital and PICU admissions in Portugal were described. Health surveillance information analysis revealed an essential and urgent need for improvement.

General characteristics: although there is accessible and exhaustive knowledge regarding adult COVID-19,[31 32] paediatric COVID-19 epidemiology and clinical characteristics are significantly less studied and documented in the literature. SARS-CoV-2 infection studies have shown that children are more likely to be asymptomatic than adults[33] and present a less severe disease course.[7 18] Young children and infants are reported as more vulnerable to infection[7] than older paediatric

patients. COVID-19-related fatalities have been reported as scarce in the paediatric population.[6] In this context, we assessed the characteristics and clinical features of COVID-19 in children in Portugal.

We observed an increase of COVID-19 cases with age in both sexes with slightly higher number of cases in males compared with females (50.5% vs 49.5%). Overall, in the scientific literature, males are most affected, although individual smaller studies have shown either sex as being the most frequent.[9 18]

Most of these cases were geographically clustered in the high-density urban areas of Lisbon, Porto and Braga. The association between population density and cumulative infection cases spread was previously observed at a county level in the USA and considered an effective predictor of cumulative infection cases.[34] The regions with lower number of cases were mainly rural and low-density populated districts, such as Portalegre, Beja and the autonomous regions of the Azores and Madeira. The nationality of paediatric patients with COVID-19 reflects the population's background. The majority were Portuguese and 4% come from diverse backgrounds such as Brazil, Angola, Cape Verde and São Tome and Principe, countries with historical ties and emigration patterns to Portugal.

Several COVID-19 waves were observed in our study. Following the first wave, in March and April 2020, the first State of Emergency,[35] and lockdown were

**Table 3** Comorbidities of the paediatric COVID-19 cases, from March 2020 to February 2021, in Portugal

| | Total | Female | Male |
|---|---|---|---|
| | N (%)* | N (%)† | N (%)† |
| Total cases with comorbidities‡ | 1593 | 691 | 902 |
| Total cases with known comorbidities | 1000 | 427 | 573 |
| Number of comorbidities | | | |
| 0 | 18 087 (94.9%) | 9013 (95.5%) | 9074 (94.1%) |
| 1 | 943 (4.9%) | 401 (4.2%) | 542 (6.0%) |
| 2 | 47 (0.2%) | 20 (0.2%) | 27 (0.3%) |
| 3+ | 10 (0.1%) | 6 (0.1%) | 4 (0.0%) |
| Total | 19 087 | 9440 | 9647 |
| Comorbidities | | | |
| Asthma | 753 (47.3%) | 316 (45.7%) | 437 (48.4%) |
| Chronic neurological/neuromuscular disease | 49 (3.1%) | 24 (3.5%) | 25 (2.8%) |
| Diabetes | 55 (3.5%) | 29 (4.2%) | 26 (2.9%) |
| Chronic haematological disease | 40 (2.1%) | 17 (2.5%) | 23 (2.5%) |
| Chronic neurological deficit | 34 (2.1%) | 15 (2.2%) | 19 (2.0%) |
| Cancer | 36 (2.3%) | 13 (1.9%) | 23 (2.5%) |
| Chronic pulmonary disease | 42 (2.6%) | 17 (2.5%) | 25 (2.8%) |
| Chronic renal disease | 28 (1.8%) | 15 (2.2%) | 13 (1.4%) |
| HIV and other immunodeficiencies | 25 (1.6%) | 12 (1.7%) | 13 (1.4%) |
| Liver disease | 8 (0.5%) | 2 (0.3%) | 6 (0.6%) |

Results of each morbidity were presented considering the sample size of cases with any reported morbidity (n=1593).
*Global percentage.
†Percentage within sex level.
‡Total cases with comorbidities, of which 1000 are specified.

implemented, including school closures, mandatory remote work, limiting mobility, business hours and social gatherings. Between 17 March and 2 April, all children who attended the emergency department (ED) of a Portuguese hospital with suspected COVID-19 symptoms as cough, fever or rhinorrhoea were tested. Of all 94 cases, 1 was positive.[36] This can indicate a low incidence of disease at this stage, which is in accordance with our results. Studies on antibody seroprevalence of SARS-CoV-2 have shown that only half the children and adolescents had developed disease symptoms.[37]

Clinical presentation and comorbidities: the most common symptoms identified (cough, fever, rhinorrhoea and headache) have been dominant in meta-analysis describing clinical characteristics of children. Wang *et al* reported the presence of fever in 48% and cough in 39% of the cases.[38] Li *et al* in 7004 paediatric cases reported fever 47% and cough 42% as the most prevalent clinical symptoms.[39] The presence of gastrointestinal symptoms has been identified as major symptoms occurring in the paediatric population.[40–42] In our analysis, diarrhoea, abdominal pain, nausea and vomiting were less frequent than other symptoms like cough, fever and rhinorrhoea as has been reported in other studies.[4 18] Interestingly, the most common

symptoms per age groups can be organised in three sets: (1) from ages 0 to 4 years: fever, cough and rhinorrhoea; (2) from ages 5 to 9 years: fever, cough, rhinorrhoea and headache and (3) from 10 to 17 years: the previous four symptoms, odynophagia and myalgia. This result can reflect simultaneously the inability of young children to express their complaints and an increase in the variety of symptoms and presentations as age increases.

Asthma was the most common underlying condition and has been the most reported comorbidity among several paediatric cases as well.[5 6 18] Verma *et al*[43] analysed SARS-CoV-2 infections in children in four hospitals in the New York City area and found those with asthma were more likely to need respiratory support during hospitalisation. In a systematic review, including 5686 children aged under 18 years with SARS-CoV-2 infection, the most frequent comorbidity was cardiac disease; and most children who required ventilation had underlying comorbidities.[4] A study of ED and admitted paediatric patients with COVID-19 at a Level 3 Lisbon Hospital, one of two reference hospitals for paediatric patients with COVID-19 at this point in the pandemic, showed 10% had risk factors for disease, 20% were asymptomatic, 43% presented with

**Table 4** Paediatric hospital from March 2020 to December 2020 and PICU admissions from March 2020 to February 2021 in Portugal

**Hospital admissions from March to December 2020**

| Age group | Hospital admissions n (%) | Cases per age group | Hospital admissions per 100 reported cases | Hospital admissions per 10 .000 children |
|---|---|---|---|---|
| 0–4 | 228 (47.9%) | 10 161 | 2.24 | 5.2 |
| 5–11 | 112 (23.5%) | 18 358 | 0.61 | 1.7 |
| 12–17 | 136 (28.6%) | 22 674 | 0.60 | 2.2 |
| Total | 476 (100%) | 51 193 | 0.93 | 2.8 |

**PICU admissions from March 2020 to February 2021**

| | Severe COVID-19 | MIS-C | Total ICU admissions |
|---|---|---|---|
| PICU admissions (%) | 4 (12.5%) | 28 (87.5%) | 32 (100%) |
| PICU admissions per 10 000 reported cases | 0.43 | 3.04 | 3.48 |
| PICU admissions per 10 000 children | 0.02 | 0.16 | 0.19 |
| Female (%) | 3 (75%) | 7 (25%) | 10 (31.2%) |
| Male (%) | 1 (25%) | 21 (75%) | 22 (68.8%) |
| Minimum age (months) | 3 | 15 | |
| Maximum age (years) | 17 | 17 | |
| Median (years) | 0 | 10.5 | |
| Comorbidities present (%) | 4 (100%) | 6 (21.4%) | 10 (31.2%) |

MIS-C, multisystem inflammatory disease in children; PICU, paediatric intensive care unit.

fever and 42% with respiratory symptoms.[44] Although the referred study is likely representing the most symptomatic COVID-19 cases seeking medical help, our study focuses on all paediatric COVID-19 cases, chronic respiratory disease was also the most common comorbidity. Only a small proportion of children with COVID-19 had registered comorbidities (n=1593, 2%), which is lower than reported in other studies[18] and of what the authors expected.

Given the abnormally high number of missing clinical presentations (48%, n=44 177) and comorbidities (98%, n=72 371), the quality of surveillance data was assessed. Statistical significant data results progressively worsened with the pandemic and particularly when case numbers sharply increased. These results can be consequential of massive work overload observed in the public health units combined with short staff. Furthermore, with increasing case counts, non-public healthcare professionals were recruited to support the units. However, significant asymmetries were also evident according to district, with substantial fluctuations in recording rates across the country. The two districts with the highest recorded data were Lisbon and Faro, whereas the lowest recorded on disease presentation were Guarda and Bragança and on the presence of comorbidities were Évora, Bragança, Guarda and Portalegre. Notwithstanding, good practice and the SINAVE guidelines warrant all fields to be answered and to achieve this goal measures and

geographically adapted strategies, such as increased training, resources and continued auditing, need to be implemented to improve data surveillance. The SINAVE system collects symptoms and comorbidities in a non-mandatory and non-intelligent way, not tailored to previous answers. This system also needs to be improved from a user perspective so to fast-track user experience and enhance clinical record keeping. An opportunity for improvement in the Portuguese surveillance data collection system needs to be recognised, so it can be improved with increased training, geographic adapted strategies and information systems to facilitate adequate epidemiological surveillance collection. These data are essential to guide public health strategies, policy decision making, audit and research.

Hospital and PICU admissions: in 2020, a total of 476 children were admitted to the hospital from 51 153 cases reported, corresponding to 1 in 108 cases requiring hospitalisation, a rate of 0.93% of cases nationwide and a paediatric hospitalisation rate of 2.8 per 10 000 children and adolescents. A lower case hospitalisation result than reported in neighbouring Navarra, Spain (3.6%) during the first wave,[45] in the USA (5.3%)[46] and in Scotland, where 1.03% of cases in children aged 5–17 years required hospitalisation.[47] These results might reflect a higher testing rate of asymptomatic or mildly ill children in Portugal.

Thirty-two cases required PICU admission, corresponding to a rate of admission of 3.48 per 10 000 cases. Four cases (12.5%) were admitted with severe COVID-19 (non-MIS-C) and 28 (87.5%) with MIS-C, implying that in our population, severe COVID-19 in children is less frequent than MIS-C. Patients with severe COVID-19 were predominantly female under 1 with comorbidities, whereas MIS-C patients were predominantly male and the majority (75%) had no comorbidities, which is similar to previously published results, where most MIS-C patients were healthy, and no correlation was observed between comorbidity score and MIS-C occurrence.[29]

Deaths: during this period, the DGS confirmed two paediatric deaths caused by COVID-19, corresponding to a case fatality rate of approximately 1.8 per 100 000 cases. Furthermore, analysing specifically total paediatric deaths in Portugal in 2020 (n=387) and comparing with the paediatric COVID-19 deaths, COVID-19 was the main cause of death in 0.52%. The mortality rate due to COVID-19 was 1.2 per million children and adolescents. These results find parallel to similar studies, highlighting the low mortality and case fatality rates. Despite the extremely small number of deaths, it is reasonable to compare these results with other countries. Bhopal et al reported an overall mortality rate due to COVID-19 in children of 1.7 per 10 000, until February 2021 when comparing seven countries. Smith et al[48] reported a mortality rate due to COVID-19 of 2 per million and fatality rate of 5 per 100 000 in England, between March 2020 and February 2021, a result 1.7 and 2.5 times higher than the one we reported, respectively. Spanish mortality results were published by Tagarro et al[49] and reported 0.21 per 100 000 in children aged 0–9 years and 0.34 per 100 000 in children aged 10–19 years. These results may reflect differences in shielding of vulnerable population or lockdown policies implemented in these countries, but further studies are required to analyse these effects.

The major strengths of our study are the combined analysis of a large database of COVID-19 cases, hospital and PICU admissions and deaths. These results present a description of paediatric COVID-19 in Portugal and allow comparison with other countries.

The authors found the high level of missingness in surveillance data unusual. The analysis of real-world data has important challenges as previously acknowledged,[50] however other studies using national data have reported similar difficulties.[51 52] Despite the authors' best effort in this analysis, less than optimal real-world data have imposed important limitations to the study and hindered some of its quality, particularly relating to missing values in some of the analysed variables. The authors are developing further studies and proposing solutions to help prevent these issues.[51 52] As stated, the major limitations of this analysis are data-related, the possible under-reporting of cases and its characteristics, such as symptoms and comorbidities, the limited

access to hospital admission data and deaths and the impracticality of establishing a chronological order of events. Furthermore, the pediatric mid 2020 population was used by proxy as an estimate population for the period analyzed.

## CONCLUSIONS

The overall reported case incidence was 5.4 per 100 children and adolescents and <1% of cases required hospital admission. MIS-C was more frequent in patients with no comorbidities and males, severe COVID-19 was rarer and occurred mainly in children under 1 year of age and females, all with comorbidities. Case fatality rates and mortality rates were low when comparing with other countries. This study highlights the need to improve epidemiological surveillance data in Portugal, with geographic adapted strategies, and information systems to facilitate adequate epidemiological surveillance data quality (online supplemental file 1).

**Author affiliations**
[1]EPI Task-Force FMUL, Faculdade de Medicina, Universidade de Lisboa, Lisboa, Portugal
[2]Unidade de Saúde Pública Francisco George, Lisboa, Portugal
[3]Área Disciplinar Autónoma de Bioestatística (Laboratório de Biomatemática), Faculdade de Medicina, Universidade de Lisboa, Lisboa, Portugal
[4]Instituto de Saúde Ambiental, Faculdade de Medicina, Universidade de Lisboa, Lisboa, Portugal
[5]Laboratório Associaodo TERRA, Faculdade de Medicina, Universidade de Lisboa, Lisboa, Portugal
[6]Clinica Universitaria Estomatologia, Faculdade de Medicina, Universidade de Lisboa, Lisboa, Portugal
[7]CIDNUR - Centro de Investigação, Inovação e Desenvolvimento em Enfermagem de Lisboa, Escola Superior de Enfermagem de Lisboa, Lisboa, Potugal
[8]CRC-W—Católica Research Centre for Psychological, Family and Social Wellbeing, Universidade Católica Portuguesa, Lisboa, Portugal
[9]Instituto de Medicina Preventiva e Saúde Pública, Faculdade de Medicina, Universidade de Lisboa, Lisboa, Portugal
[10]Instituto Nacional de Saúde Doutor Ricardo Jorge, Lisboa, Portugal
[11]NOVA National School of Public Health, Public Health Research Centre, Universidade NOVA de Lisboa; Comprehensive Health Research Center (CHRC), Lisbon, Portugal

**Acknowledgements** The authors would like to thank Dr Cristina Camilo and the EPICENTRE.PT research group.

**Contributors** CE planned the study and was the guarantor. She developed the methodology and was involved in obtaining the data, data analysis, writing the manuscript, creating tables/figures and reviewing the text. RF-S was involved in obtaining data, writing the manuscript, creating tables and figures and reviewing the text. CC was involved in planning the study, creating tables and figures, writing and reviewing the manuscript and maintaining references. MdAN was involved in writing and reviewing the manuscript. AC was involved in writing and reviewing the manuscript. LB was involved in writing and reviewing the manuscript. CF was involved in writing and reviewing the manuscript. PN was involved in planning the study, data analysis and writing and reviewing the manuscript.

**Funding** Funds from Fundação para a Ciência e a Tecnologia to the Instituto de Saúde Ambiental, Faculdade de Medicina, Universidade de Lisboa (UIDB/04295/2020).

**Map disclaimer** The inclusion of any map (including the depiction of any boundaries therein), or of any geographic or locational reference, does not imply the expression of any opinion whatsoever on the part of BMJ concerning the legal status of any country, territory, jurisdiction or area or of its authorities. Any such

expression remains solely that of the relevant source and is not endorsed by BMJ. Maps are provided without any warranty of any kind, either express or implied.

**Competing interests**  None declared.

**Patient and public involvement**  Patients and/or the public were not involved in the design, or conduct, or reporting, or dissemination plans of this research.

**Patient consent for publication**  Not applicable.

**Ethics approval**  This study involves human participants but was not approved by Epidemiological study involving anonymous database.

**Provenance and peer review**  Not commissioned; externally peer reviewed.

**Data availability statement**  Data may be obtained from a third party and are not publicly available.

**ORCID iDs**
Cecilia Elias http://orcid.org/0000-0002-3922-7152
Rodrigo Feteira-Santos http://orcid.org/0000-0002-5780-2288
Miguel de Araújo Nobre http://orcid.org/0000-0002-7084-8301
Andreia Silva Costa http://orcid.org/0000-0002-2727-4402
Leonor Bacelar-Nicolau http://orcid.org/0000-0003-0421-1262
Cristina Furtado http://orcid.org/0000-0003-2632-0945
Paulo Jorge Nogueira http://orcid.org/0000-0001-8316-5035

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
