## [Reviewer comments · BMJ Paediatrics Open]

ARTICLE DETAILS

TITLE (PROVISIONAL)	COVID-19 in Portugal: a retrospective review of paediatric cases, hospital and PICU admissions in the first pandemic year
AUTHORS	Elias, Cecilia Feteira-Santos, Rodrigo Camarinha, Catarina de Araújo Nobre, Miguel Costa, Andreia Silva Bacelar-Nicolau, Leonor Furtado, Cristina Nogueira, Paulo Jorge

VERSION 1 – REVIEW

REVIEWER	Reviewer name: Dr. Elizabeth B. Pathak Institution and Country: Womens Inst Independent Social Enquiry, United States Competing interests: None
REVIEW RETURNED	22-Apr-2022

GENERAL COMMENTS	The topic of this paper, the descriptive epidemiology of COVID-19 in children, is very important, and national-level studies such as this manuscript are needed. Unfortunately, several critical methodological areas are significantly underdeveloped in this manuscript and prevent me from recommending acceptance. 1. Missing Data The majority of the case records in these national surveillance data are missing information about the presence or absence of specific symptoms. While this fact is stated in the paper, the implications for analysis and interpretation of findings are not sufficiently appreciated by the authors. For example, the finding that children with comorbidities were more likely to have symptoms reported could have easily resulted from reporting bias, not from a true underlying difference in the population, and the authors have no information to allow them to rule out bias as an explanation for their findings. Similarly, for comorbidities, we are told that approximately 2% of pediatric cases had a reported comorbidity. But the question of whether the remaining 98% of cases did NOT have comorbidities vs. were missing information about comorbidities is unanswered in the paper. The high level of missingness should entail very cautious interpretation of associations and causation. Because missingness across multiple variables in a data record is not random, spurious associations can emerge. Pediatric patients who were not missing comorbidity data would also be much more likely to NOT be missing symptom data.
--

Furthermore, the authors should conduct and report a systematic analysis of the pattern of missingness by age and gender, and possibly by geographic area and time period as well. Did the quality of surveillance data reporting improve over time? Was it higher or lower in major cities vs. rural areas?

2. Case counts vs. incidence

The authors conflate numerator counts of cases with "incidence." Incidence must be expressed as a rate, with an appropriate census-derived population denominator. This is a very elementary error. The authors describe incidence increasing by single years of age in the text, when the data only show increasing numerator counts - there are no incidence rates presented anywhere in the paper. Similarly, the map and discussion of geographic patterns is based on numerators only. Obviously there will always be a larger number of cases in cities than in rural areas. The authors have not investigated whether the incidence RATES are higher in urban vs. rural areas.

3. Hospitalization and mortality analyses and discussion

These data are not contextualized at all. How complete was the reporting of pediatric hospitalizations? Were these data directly linked to the case surveillance data, or did they come from an independent, unlinked data source? Is child/family socioeconomic position a barrier to hospitalization in Portugal? Does access to medical care vary geographically?

Hospitalization rates should be presented two ways - population rates (with the child population at risk as the denominator) and case rates (with the child cases as the denominator).

Infants should be separated from the 0-4 group, because both hospitalization and death rates are known to be much higher for infants than for children 1-4 years old.

The quality of the death data should be described and discussed - what is known about out-of-hospital COVID-19 mortality, misclassification of cause of death (especially in the early days of the pandemic), and any urban-rural differences in quality of mortality surveillance?

4. Data Tables

As stated above, many of these tables should be reformulated with incidence rates, not just numerator case counts.

In Table 1, cases who were missing gender need to be accounted for in the table.

In Table 2, the total number of cases in each age group should be included in the top header row of the table.

In Table 3, the percents reported in the Female and Male columns are "row percents" and they should be "column percents" to match the way the percents are calculated in the Total column. Also include the totals for females and males in the header row.

Supplemental Table 2 is very misleading. The majority of cases with missing symptom reports are not included here. The conservative approach would be to assume those cases were asymptomatic - at the very least they should be included as a separate column. The percents as currently shown are uninformative. They should be replaced with row percents, to answer the question "What percent of cases at this year of age had symptoms reported in the data system?"

REVIEWER	Reviewer name: Dr. Harish PEMDE Institution and Country: Kalawati Saran Children's Hospital, India Competing interests: None
REVIEW RETURNED	25-Apr-2022

GENERAL COMMENTS	This is an important documentation. However, no information was available for a large number of patients (>62%). These patients have also been included in denominator for calculating the percentage of patients having various symptoms. This is not appropriate. Some recalculations are needed. More information about patients with MISC and death would be better. In Table-2, provide total number of patients studied in each age group. In Table-3, the calculations of percentages are confusing. For example, the number of co-morbidities under the column Total are 2% but it becomes 42.6% for Females and 57% for Males.
--

REVIEWER	Reviewer name: Dr. Mariana Poppe Institution and Country: Hospital Beatriz Angelo, Portugal Competing interests: None
REVIEW RETURNED	21-Apr-2022

GENERAL COMMENTS	GENERAL COMMENT: The theme of the article is very relevant. Publication of national data is helpful to understand the "bigger picture" of this pandemic and might help to ground future measures on solid scientific evidence. The title reflects the content of the article appropriately. The abstract is well-constructed, needing small adjustments. The background section correctly presents the context of the study. The objectives are clearly defined. More information concerning the methods and analysis is required for better understanding and interpretation of the results of the study. Further statistical analysis could enhance the quality of the study. The presentation of the results could be improved in order to be coherent with the abstract and discussion. A major limitation concerning the study results hinders the conclusions of the article. The discussion is well written, although with a few statements without the appropriate grounding that need to be revised. The limitations of the study are portrayed. The conclusion responds to the proposed objectives, needing small adjustments. ABSTRACT - Page 2, line 16: The time period to which the study refers is not mentioned, it could enrich the abstract to include that information. - Page 4, line 19: it reads "dyspnoea", while in the abstract the authors used "dyspnea" – the authors should use one or the other spelling coherently throughout the article. BACKGROUND
--

- Page 5, lines 27-29: "the role of the pediatric population in SARS-CoV-2 spread in the community have been considered of major interest" – is this worth mentioning in the background, since your research does not approach this any further nor is it ever mentioned again during the article?

- Page 5, lines 33-35: "A small number of children need hospital admission, and of these, a minority required ICU admission either due to severe COVID-19 or multisystem inflammatory disease in children (MIS-C)" – the verb 'need' is in the present, the verb 'required' in the past tense.

MATERIALS AND METHODS

- Page 6, line 14: It is relevant for non-Portuguese readers (or Portuguese non-health workers) to clarify what information is collected through SINAVE and how it is collected. The article does not make clear that health workers are the ones who fill out the form and what data is included. This would be important to understand the limitations of the study further ahead in the article.

- Page 6, line 21: The article does not mention the time frame for data on PICU admissions. Since there are different time frames for the diagnosis of COVID-19 and for hospital admissions, it is important to make this information clear.

- Page 6, line 23: The article does not disclose the definition used in the study for severe COVID-19 nor MIS-C. This information is relevant, either explained in the article (as done with the definition of a confirmed COVID-19 case) or with use of a reference.

RESULTS

- Page 6, line 56: "The present study included 92051 COVID-19 cases in children and adolescents reported from March 2nd, 2020, to February 28th, 2021, as infected with SARS-CoV-2 in Portugal;"
– The term 'COVID-19 cases' already states that they are 'infected with SARS-CoV2' as defined in the methods, no need to repeat the information.

- Page 7, line 11: "Porto 18350 (19.9%)" – Either the absolute number is written inside the brackets ($n = 18350, 19.9\%$), or it should be included in the sentence, like 'Porto with 18350 cases'. The same with Braga.

- Page 7, lines 27-35: "COVID-19 CASES TIMELINE" – The article only specifies 'lockdown easing' in the 4th epidemic wave. I would suggest that the authors either characterize the national measures (lockdown, school closure/opening, etc) for every wave or for none. Only characterizing one of them gives the reader only partial information. This could be left for the discussion section if intended by the authors.

- Page 7, line 47: The major limitation of the study is portrayed here – 62.5% of cases without information regarding symptoms. This limitation is utterly relevant, as it has the potential to hinder all the other conclusions of the study concerning the presenting symptoms – one could argue that the article only analyses 37.5% of cases in Portugal when approaching symptoms. It could be that gastrointestinal symptoms were predominant in the two thirds of patients without reported symptoms, or that any other information collected on such a large proportion of the sample could change the conclusions of this study. Given the relevance of this matter, one could argue that the analysis of the study should focus only on the data from completely filled-out forms on the SINAVE platform, excluding the incomplete forms, to avoid misleading interpretations. I stand with great doubt about whether this is

information should be disclosed right from the start in the abstract to avoid misleading the reader.

- Page 7, line 55: "(10205, 11.1%)," – should read: '(n = 10205, 11.1%)', as it reads in the paragraph concerning comorbidities.

- Page 7, line 40: CLINICAL PRESENTATION.
The gastrointestinal symptoms are barely referred in the results, however they are enhanced in the abstract. If they are mentioned in the 'results' part of the abstract ("Gastrointestinal symptoms were infrequent."), this information should be contained in the results section of the article.

- Page 8, line 20: COMORBIDITIES. Were the comorbidities an obligatory answer while filling out the SINAVE form, or is it possible that they are also under-reported due to incomplete SINAVE forms?

- Page 8, line 37: "Patients with comorbidities presented symptoms more frequently than patients with no comorbidities." Is there a statistically significant difference? Were comparison tests performed?

- Page 9, line 24: "Two deaths caused by COVID-19 were reported by the DGS during this study." These deaths were not in PICU admitted patients?

DISCUSSION

- Page 10, line 34: "significant differences between sex were observed mainly between 16 and 17, and predominantly in females." This information belongs in the results section, it was not mentioned there. 'Significant differences' implies that comparison tests were performed, but they are not mentioned. Or is it based only on the information in table 1 (50.9% vs 49.1% for 16 years and 50.7% vs 49.3% for 17 years)?

- Page 10, line 41: "geographic" should read 'geographically'.

- Page 10, line 47: "Though" does not make sense within the sentence, revise the phrase.

- Page 10, line 52: "diverse backgrounds Brazil, Angola" should read 'diverse backgrounds such as Brazil,' ... or the names of countries in brackets.

- Page 10, last paragraph: When referring to the study's results, verbs are used in the present tense (are, reflects, etc) and in the past tense (were) – revise throughout the article.

- Page 10, line 57: "the pediatric and adult population peaks co-occurred simultaneously". Lacking a reference for this statement.

- Page 11, line 23: "such has been reported elsewhere" – 'as has been reported in other studies'.

- Page 11, line 31: "As found in our results (38.1%), the combination of symptoms, fever and cough 30% has also been documented." Revise formulation of the sentence, hard to understand.

- Page 11, line 43: "Unknown clinical presentations were frequent and while the majority of these are likely to be asymptomatic patients in which no symptoms were denied on the digital platform SINAVE". Based on which grounds (or references) can the authors state that these are likely asymptomatic patients? One could claim

that is a possibility, but not affirm it without objective data supporting the affirmation. This is the main limitation of the study, and thus should be portrayed objectively.

Once more, this evidences why characterizing the SINAVE platform in the methods section is important, otherwise readers will not be able to fully understand what is being discussed in this section.

- Page 12, line 23: "Lisbon Hospital". Clarifying it is a level 3 hospital and one of the two reference centres for paediatric Covid-19 patients during the early stage of the pandemic might shed light on the high percentage of risk factors found.

- Page 12, line 47: "admitted to hospital" – 'admitted to the hospital'.

- Page 13, line 11: "as less frequent than" – 'is less frequent than'

- Page 13, line 52: "The authors found unusual the lack of surveillance data in the analyzed dataset" – 'The authors found the lack of surveillance data in the analysed dataset unusual'

CONCLUSIONS

- Page 14, line 23: "analyses" – 'analysis'.

- Page 14, line 32: "severe COVID-19 was rarer occurred mainly" – 'severe COVID-19 was rarer and occurred mainly'.

- Page 14, line 35: "Case fatality rates and mortality rates were extremely low when comparing with other countries". The authors compared the fatality rate of the study with data from England (which was higher) and Spain (which was lower) – thereby this conclusion is not accurate.

- The improvement in the reporting system could also be mentioned in the conclusions section, being an important report of this study.

TABELES AND FIGURES

- TABLE 1. AGE AND SEX OF PEDIATRIC COVID-19 INFECTED CASES, DURING 2020 IN PORTUGAL. – It is not accurate to state these are the cases 'during 2020', since the time frame is March 2020 to February 2021. The word 'infected' is unnecessary.

- TABLE 3. COMORBIDITIES OF THE PEDIATRIC COVID-19 INFECTED CASES, DURING 2020 IN PORTUGAL. – It is not accurate to state these are the cases 'during 2020', since the time frame is March 2020 to February 2021. Footnotes should include description of the meaning of "***" and "****"

- TABLE 4. PEDIATRIC HOSPITAL AND PICU ADMISSIONS. In the lower part of the table, "Reported cases" should specify that it means COVID-19 cases and not reported cases of severe COVID. "Comorbidities" means having comorbidities or not having them? Maybe clarify by adding 'comorbidities present', to avoid confusion. Add 'n (%)'.

- SUPPLEMENTARY TABLE 1. PEDIATRIC COVID-19 CASES COUNTRY OF ORIGIN, DURING 2020 IN PORTUGAL. – The title of the table should refer that these are the countries of origin excluding Portugal.

- SUPPLEMENTARY TABLE 4.

Wouldn't it be more accurate to compare patients with comorbidities with patients without comorbidities, instead of all

	patients? - SUPPLEMENTARY TABLE 5. NUMBER OF SYMPTOMS REPORTED BY PEDIATRIC COVID-19 CASES WITH AND WITHOUT COMORBIDITIES Once more, the title claims to compare patients with and without comorbidities, but then the table compares comorbidities with all cases.
--	--

VERSION 1 – AUTHOR RESPONSE

Dear Editor-in-Chief of the BMJ Paediatrics Open, We now resubmit the original manuscript entitled “COVID-19 in Portugal: a retrospective review of paediatric cases, hospital and PICU admissions in the first pandemic year” to be considered for publication in the BMJ Paediatrics Open. Firstly, the authors would like to thank the editor-in-chief and the reviewers for their assessment and recommendations regarding our manuscript. Paediatric COVID-19 national studies and results are extremely relevant to fully characterize and preserve our knowledge of this pandemic. The authors set out with very specific objectives to characterize this disease in children and to compare our results with those from other countries. However, while trying to answer these questions, others rose, connected to data quality of two variables we had extensively analyzed. At the time of submission, the authors did not fully appreciate this data limitation. However, we have taken upon reflection all the reviews received and have updated our manuscript accordingly. In particular, the authors have curtailed their analysis, comparisons and conclusions connected to these variables. We removed comparisons on disease presentation and the presence of comorbidities and removed the mention of comorbidity from the section "What this study adds". Furthermore, without taking the focus of our study away from a paediatric COVID-19 case description and analysis, and as recommended by one of the reviewers, we have conducted a missingness analysis. This analysis can prove an important tool to improve surveillance data in our country. Major changes affecting all sections and tables of the manuscript have been taken in-keeping with peer-review suggestions. Given the lengthy revision, the authors highlighted revised segments rather than add line-by-line tracking. Point-by-point answers to the reviewers are provided in separate word documents. We reaffirm this work has neither been published nor is currently submitted for publication elsewhere in whole or in part and all the listed authors have contributed significantly to this work and approved it. Altogether, we are confident that our contribution is suitable for publication in the BMJ Paediatrics Open, and we look forward to hearing back in due course. Kind regards, Cecília Elias Universidade de Lisboa, Faculdade de Medicina Instituto de Medicina Preventiva e Saúde Pública Avenida Professor Egas Moniz, 1649-028 Lisboa, Portugal e-mail: cecilia.elias@gmail.com

REVIEWER 1 GENERAL - [Comment 1] The theme of the article is very relevant. Publication of national data is helpful to understand the “bigger picture” of this pandemic and might help to ground future measures on solid scientific evidence. Response: The authors would like to thank the reviewer for recognizing the relevance of this study. National studies portraying paediatric COVID-19 patients are indeed important and can help frame this pandemic for future knowledge and as basis for measures and evidence-based policy making. - [Comment 2] The title reflects the content of the article appropriately. The abstract is wellconstructed, needing small adjustments. The background

section correctly presents the context of the study. The objectives are clearly defined. Response: Thank you for the kind comments. The abstract, background and objectives have been reviewed in accordance with peer-review suggestions. - [Comment 3] More information concerning the methods and analysis is required for better understanding and interpretation of the results of the study. Further statistical analysis could enhance the quality of the study. Response: The authors have updated all sections of the manuscript. As suggested, the authors have added information to the methods section and, given the high level of missingness on two variables extensively studied in the previous version, have tailored their analysis, limiting comparisons and conclusions from these variables, in accordance with reviewers. A missingness analysis was similarly conducted. - [Comment 4] The presentation of the results could be improved in order to be coherent with the abstract and discussion. A major limitation concerning the study results hinders the conclusions of the article. Response: The authors reviewed the abstract and discussion and added information to the results section, which was lacking. We reviewed our major limitation, associated with the high level of missingness on 2 variables studied in our analysis, and decided to scaleback our comparisons from these variables. Concomitantly, we added a missingness analysis which provides an insight on the causes and potential improvements to be achieved with the reporting system. Nevertheless, the authors tried to maintain their main goal to describe pediatric COVID-19 in Portugal. - [Comment 5] The discussion is well written, although with a few statements without the appropriate grounding that need to be revised. The limitations of the study are portrayed. 2 Response: The authors have extensively revised the discussion and segments needing referencing were edited in accordance. In particular, the discussion on missing epidemiological surveillance data was replaced with the discussion of the missingness analysis. - [Comment 6] The conclusion responds to the proposed objectives, needing small adjustments. Response: The authors have enhanced the conclusion section considering the need to improve surveillance data. ABSTRACT - [Comment 7] Page 2, line 16: The time period to which the study refers is not mentioned, it could enrich the abstract to include that information. Response: The authors would like to thank the reviewer for the suggestion and have updated the methods section of the abstract accordingly. - [Comment 8] Page 4, line 19: it reads "dyspnoea", while in the abstract the authors used "dyspnea" – the authors should use one or the other spelling coherently throughout the article. Response: The authors would like to thank the reviewer for the correction. Although this wording has been removed from the abstract, a consistent spelling throughout the manuscript has been attempted. BACKGROUND - [Comment 9] Page 5, lines 27-29: "the role of the pediatric population in SARS-CoV-2 spread in the community have been considered of major interest" – is this worth mentioning in the background, since your research does not approach this any further nor is it ever mentioned again during the article? Response: The authors would like to thank the reviewer for this comment, with which we agree, and have removed this sentence from the background section accordingly. - [Comment 10] Page 5, lines 33-35: "A small number of children need hospital admission, and of these, a minority required ICU admission either due to severe COVID-19 or multisystem inflammatory disease in children (MIS-C)" – the verb 'need' is in the present, the verb 'required' in the past tense. Response: The authors would like to thank the reviewer for this comment and have revised the sentence accordingly. 3 METHODS - [Comment 11] Page 6, line 14: It is relevant for non-Portuguese readers (or Portuguese nonhealth workers) to clarify what information is collected through SINAVE and how it is collected. The article does not make clear that health workers are the ones who fill out the form and what data is included. This would be important to understand the limitations of the study further ahead in the article. Response: The authors would like to thank the reviewer for this comment and have added

information regarding the SINAVE platform to the methods and the discussion sections. One of the simplest ways to improve health surveillance in Portugal would be to improve the SINAVE platform as presented in the discussion section, partnered with training and resources. - [Comment 12] Page 6, line 21: The article does not mention the time frame for data on PICU admissions. Since there are different time frames for the diagnosis of COVID-19 and for hospital admissions, it is important to make this information clear. Response: The authors would like to thank the reviewer for this comment and have added the timeframe for PICU admissions to the methods section. This period is different from the others analyzed due to data access constraints. - [Comment 13] Page 6, line 23: The article does not disclose the definition used in the study for severe COVID-19 nor MIS-C. This information is relevant, either explained in the article (as done with the definition of a confirmed COVID-19 case) or with use of a reference. Response: The authors would like to thank the reviewer for this comment and have added references for the definitions of severe COVID-19 and MIS-C.

RESULTS - [Comment 14] Page 6, line 56: "The present study included 92051 COVID-19 cases in children and adolescents reported from March 2nd, 2020, to February 28th, 2021, as infected with SARS-CoV-2 in Portugal;" – The term 'COVID-19 cases' already states that they are 'infected with SARS-CoV2' as defined in the methods, no need to repeat the information. Response: The authors would like to thank the reviewer for this comment and have updated this sentence as suggested. - [Comment 15] Page 7, line 11: "Porto 18350 (19.9%)" – Either the absolute number is written inside the brackets (n = 18350, 19.9%), or it should be included in the sentence, like 'Porto with 18350 cases'. The same with Braga. 4 Response: The authors would like to thank the reviewer for this comment and have updated the way absolute numbers are written in the text, as suggested. - [Comment 16] Page 7, lines 27-35: "COVID-19 CASES TIMELINE" – The article only specifies 'lockdown easing' in the 4th epidemic wave. I would suggest that the authors either characterize the national measures (lockdown, school closure/opening, etc) for every wave or for none. Only characterizing one of them gives the reader only partial information. This could be left for the discussion section if intended by the authors. Response: The authors would like to apologize for this inattention. In a previous version of the manuscript there was an extensive description of COVID-19 waves and implemented public health social measures that was removed but some of the text referring to it was not updated. - [Comment 17] Page 7, line 47: The major limitation of the study is portrayed here – 62.5% of cases without information regarding symptoms. This limitation is utterly relevant, as it has the potential to hinder all the other conclusions of the study concerning the presenting symptoms – one could argue that the article only analyses 37.5% of cases in Portugal when approaching symptoms. It could be that gastrointestinal symptoms were predominant in the two thirds of patients without reported symptoms, or that any other information collected on such a large proportion of the sample could change the conclusions of this study. Given the relevance of this matter, one could argue that the analysis of the study should focus only on the data from completely filled-out forms on the SINAVE platform, excluding the incomplete forms, to avoid misleading interpretations. I stand with great doubt about whether this information should be disclosed right from the start in the abstract to avoid misleading the reader. Response: As stated previously, by the authors, the high level of missingness in our analysis is an important constraint. To minimize these limitations the authors have curtailed their analysis, comparisons and conclusions regarding disease presentation and the presence of comorbidity variables. We reviewed our analysis to exclude the missing cases from percentages calculi. Furthermore, without taking the focus of our study away from a paediatric COVID-19 case description and analysis we have conducted a missingness analysis. Information regarding the level of missingness has been included in the

abstract and also in the beginning of the discussion. The authors were able to reduce missing information by recoding the disease presentation and comorbidities variables, as described in the method section. - [Comment 18] Page 7, line 55: "(10205, 11.1%)" – should read: '(n = 10205, 11.1%)', as it reads in the paragraph concerning comorbidities. Response: The authors would like to thank the reviewer for this comment and have updated the manuscript accordingly. 5 - [Comment 19] Page 7, line 40: CLINICAL PRESENTATION. The gastrointestinal symptoms are barely referred in the results, however they are enhanced in the abstract. If they are mentioned in the 'results' part of the abstract ("Gastrointestinal symptoms were infrequent."), this information should be contained in the results section of the article. Response: The authors would like to thank the reviewer for this comment and recognize gastrointestinal symptoms were lacking in the results section. These have been updated in the manuscript as suggested. - [Comment 20] Page 8, line 20: COMORBIDITIES. Were the comorbidities an obligatory answer while filling out the SINAVE form, or is it possible that they are also under-reported due to incomplete SINAVE forms? Response: The authors would like to thank the reviewer for this pertinent comment, as stated previously, the high level of missingness in the disease presentation and the presence of comorbidities variables is a major limitation which the authors did not fully appraise initially. None of these variables are mandatory in the SINAVE collected information. Comorbidities were reported (as present or absent) in 19,680 cases (21.4%) which is clearly inadequate. From our missingness analysis, reported comorbidity information decreased as the pandemic progressed and varied between districts and sexes. In light of this, the authors excluded missing information from our percentages and abstained from comparing results obtained from these variables. Information forewarning to the level of missingness has been included in the abstract and also at the beginning of the discussion. - [Comment 21] Page 8, line 37: "Patients with comorbidities presented symptoms more frequently than patients with no comorbidities." Is there a statistically significant difference? Were comparison tests performed? Response: We thank the reviewer for this comment, in light of previously discussed study limitations we opted to remove this comparison from our study. - [Comment 22] Page 9, line 24: "Two deaths caused by COVID-19 were reported by the DGS during this study." These deaths were not in PICU admitted patients? Response: We thank the reviewer for this comment, we obtained data on the two COVID-19 deaths from the SINAVE data and confirmed these with DGS death information. The authors do not have data to ascertain the location of death, but it is likely they occurred in PICU. 6 DISCUSSION - [Comment 23] Page 10, line 34: "significant differences between sex were observed mainly between 16 and 17, and predominantly in females." This information belongs in the results section, it was not mentioned there. 'Significant differences' implies that comparison tests were performed, but they are not mentioned. Or is it based only on the information in table 1 (50.9% vs 49.1% for 16 years and 50.7% vs 49.3% for 17 years)? Response: The authors would like to apologize for this inattention. This comparison was performed at a previous version of the manuscript that was removed and the text referring to it was not updated. - [Comment 24] Page 10, line 41: "geographic" should read 'geographically'. Response: The authors would like to thank the reviewer for the correction and have updated the manuscript accordingly. - [Comment 25] Page 10, line 47: "Though" does not make sense within the sentence, revise the phrase. Response: Response: The authors would like to thank the reviewer for the correction and have updated the manuscript accordingly. - [Comment 26] Page 10, line 52: "diverse backgrounds Brazil, Angola" should read 'diverse backgrounds such as Brazil,' ... or the names of countries in brackets. Response: The authors would like to thank the reviewer for the correction and have updated the manuscript accordingly. - [Comment 27] Page 10, last paragraph: When referring to the study's results, verbs are used in the

present tense (are, reflects, etc) and in the past tense (were) – revise throughout the article.

Response: The authors would like to thank the reviewer for highlighting this inconsistency and have updated the manuscript accordingly. - [Comment 28] Page 10, line 57: “the pediatric and adult population peaks co-occurred simultaneously”. Lacking a reference for this statement. Response: The authors would like to apologize for this inattention. This analysis was performed at a previous version of the manuscript that was removed and the text referring to it was not updated. 7 - [Comment 29] Page 11, line 23: “such has been reported elsewhere” – ‘as has been reported in other studies’. Response: The authors would like to thank the reviewer for the correction and have updated the manuscript accordingly. - [Comment 30] Page 11, line 31: “As found in our results (38.1%), the combination of symptoms, fever and cough 30% has also been documented.” Revise formulation of the sentence, hard to understand. Response: The authors would like to thank the reviewer for the correction, this sentence has been removed from the manuscript. - [Comment 31] Page 11, line 43: “Unknown clinical presentations were frequent and while the majority of these are likely to be asymptomatic patients in which no symptoms were denied on the digital platform SINAVE”. Based on which grounds (or references) can the authors state that these are likely asymptomatic patients? One could claim that is a possibility, but not affirm it without objective data supporting the affirmation. This is the main limitation of the study, and thus should be portrayed objectively. Once more, this evidences why characterizing the SINAVE platform in the methods section is important, otherwise readers will not be able to fully understand what is being discussed in this section. Once more, this evidences why characterizing the SINAVE platform in the methods section is important, otherwise readers will not be able to fully understand what is being discussed in this section. Response: We thank the reviewer for this comment and to better assess and analyze the missingness observed, the authors conducted a missingness analysis. This shows missing information on disease presentation and comorbidities across districts, time and sexes were not random and significant differences were observed between groups. As previously stated, missing information, not selected from the SINAVE platform, increased as the pandemic progresses and differences were observed between districts, this data was composed of negative and positive answers that were not selected. To minimize this limitation the authors have update the analysis and manuscript thoroughly. Missing cases have been removed from denominator calculations, the analysis and comparisons of these variables has been curtailed and conclusions have been scaled-back. - [Comment 32] Page 12, line 23: “Lisbon Hospital”. Clarifying it is a level 3 hospital and one of the two reference centres for paediatric Covid-19 patients during the early stage of the pandemic might shed light on the high percentage of risk factors found. Response: The authors would like to thank the reviewer for the suggestion and have updated the manuscript accordingly. 8 - [Comment 33] Page 12, line 47: “admitted to hospital” – ‘admitted to the hospital’. Response: The authors would like to thank the reviewer for the correction and have updated the manuscript, accordingly. - [Comment 34] Page 13, line 11: “as less frequent than” – ‘is less frequent than’ Response: The authors would like to thank the reviewer for the correction and have updated the manuscript accordingly. - [Comment 35] Page 13, line 52: “The authors found unusual the lack of surveillance data in the analyzed dataset” – ‘The authors found the lack of surveillance data in the analysed dataset unusual’ Response: The authors would like to thank the reviewer for the correction and have updated the manuscript accordingly. - [Comment 36] Page 14, line 23: “analyses” – ‘analysis’. Response: The authors would like to thank the reviewer for the correction and have updated the manuscript all throughout accordingly. - [Comment 37] Page 14, line 32: “severe COVID-19 was rarer occurred mainly” – ‘severe COVID-19 was rarer and occurred mainly’. Response: The authors would

like to thank the reviewer for the correction and have updated the manuscript accordingly.

CONCLUSION - [Comment 38] Page 14, line 35: "Case fatality rates and mortality rates were extremely low when comparing with other countries". The authors compared the fatality rate of the study with data from England (which was higher) and Spain (which was lower) – thereby this conclusion is not accurate. Response: We thank the reviewer for this comment, case fatality rates were only compared with the english results. Case-fatality rates in England were higher than the ones we report. Mortality rates were compared with spanish, english, US results and with combined results from 7 countries. - [Comment 39] The improvement in the reporting system could also be mentioned in the conclusions section, being an important report of this study. 9 Response: The authors would like to thank the reviewer for this suggestion and have updated the manuscript accordingly. Improving epidemiological surveillance information must be a priority and this information should be included in the conclusion section.

TABLES - [Comment 40] TABLE 1. AGE AND SEX OF PEDIATRIC COVID-19 INFECTED CASES, DURING 2020 IN PORTUGAL. – It is not accurate to state these are the cases 'during 2020', since the time frame is March 2020 to February 2021. The word 'infected' is unnecessary. Response: The authors would like to thank the reviewer for this suggestion and have updated table 1 accordingly. - [Comment 41] TABLE 3. COMORBIDITIES OF THE PEDIATRIC COVID-19 INFECTED CASES, DURING 2020 IN PORTUGAL. – It is not accurate to state these are the cases 'during 2020', since the time frame is March 2020 to February 2021. Footnotes should include description of the meaning of "***" and "****" Response: The authors would like to thank the reviewer for this suggestion and have updated table 3 accordingly. - [Comment 42] TABLE 4. PEDIATRIC HOSPITAL AND PICU ADMISSIONS. In the lower part of the table, "Reported cases" should specify that it means COVID-19 cases and not reported cases of severe COVID. "Comorbidities" means having comorbidities or not having them? Maybe clarify by adding 'comorbidities present', to avoid confusion. Add 'n (%)'. Response: The authors would like to thank the reviewer for this suggestion and have updated the table accordingly.

SUPPLEMENTARY TABLES - [Comment 43] SUPPLEMENTARY TABLE 1. PEDIATRIC COVID-19 CASES COUNTRY OF ORIGIN, DURING 2020 IN PORTUGAL. – The title of the table should refer that these are the countries of origin excluding Portugal. Response: The authors would like to thank the reviewer for this suggestion and have updated the table's title accordingly. - [Comment 44] SUPPLEMENTARY TABLE 4 - Wouldn't it be more accurate to compare patients with comorbidities with patients without comorbidities, instead of all patients? 10 Response: We would like to thank the reviewer for this comment. Considering the limitations stated previously, the authors decided to exclude this analysis from our manuscript. - [Comment 45] SUPPLEMENTARY TABLE 5. NUMBER OF SYMPTOMS REPORTED BY PEDIATRIC COVID-19 CASES WITH AND WITHOUT COMORBIDITIES Once more, the title claims to compare patients with and without comorbidities, but then the table compares comorbidities with all cases. Response: We would like to thank the reviewer for this comment. Considering the limitations stated previously, the authors decided to exclude this analysis from our manuscript

REVIEWER 2 GENERAL - [Comment 1] The topic of this paper, the descriptive epidemiology of COVID-19 in children, is very -important, and national-level studies such as this manuscript are needed. Response: The authors would like to thank the reviewer for recognizing the importance of this study. National studies portraying paediatric COVID-19 patients are indeed important and can help frame this pandemic for future knowledge and as basis for evidence-based policy making and research. - [Comment 2] Unfortunately, several critical methodological areas are significantly underdeveloped in this manuscript and prevent me from recommending acceptance. Response: The authors would

like to thank the reviewer for pointing our critical methodology flaws. Care has been taken to improve our analysis and manuscript by addressing these concerns. Our main goal was to, enthusiastically, describe paediatric COVID-19 presentations and outcomes of patients in our country. Unfortunately, the surveillance data utilized in this analysis had high levels of missingness which greatly hampered part of the analysis conducted. We did not fully appreciate the ramifications of this in our initial analysis, in particular when describing and comparing variables with high levels of missingness. We have reflected upon your comments, have partially reanalyzed the data, added the missingness analysis and have made major manuscript editing. This analysis can prove an important tool to improve surveillance data in our country. We hope this revised version is in accordance with your comments and suggestions.

1. Missing Data - [Comment 3] The majority of the case records in these national surveillance data are missing information about the presence or absence of specific symptoms. While this fact is stated in the paper, the implications for analysis and interpretation of findings are not sufficiently appreciated by the authors. For example, the finding that children with comorbidities were more likely to have symptoms reported could have easily resulted from reporting bias, not from a true underlying difference in the population, and the authors have no information to allow them to rule out bias as an explanation for their findings. Response: The authors would like to thank the reviewer for these comments which greatly impacted our assessment and analysis. The high level of missingness is a major limitation and, as you stated, interconnecting symptoms and comorbidities to obtain results from this data was not appropriate and likely introduced bias to our analysis. To minimize this limitation the authors have curtailed their analysis, comparisons and conclusions regarding disease presentation and the presence of comorbidity variables. We reviewed our analysis to exclude the missing cases from percentages calculations. Furthermore, without taking the focus of our study away from a paediatric COVID-19 case description and analysis we have conducted a missingness analysis, as you suggested. Important information was obtained and statistical differences between district and month were observed. Information regarding the level of missingness has been added throughout the manuscript and in particular in the abstract and at the beginning of the discussion. The need for improved surveillance information has also been highlighted in the abstract, discussion and conclusion. Given our study focuses on reported cases, hospital and PICU admissions and deaths with this review we have decreased our focus on the reported cases presentation and description and refocused on the other components of the analysis.

- [Comment 4] Similarly, for comorbidities, we are told that approximately 2% of paediatric cases had a reported comorbidity. But the question of whether the remaining 98% of cases did NOT have comorbidities vs. were missing information about comorbidities is unanswered in the paper. Response: The authors would like to thank the reviewer for this comment. As stated previously, the high level of missingness in the disease presentation and in the presence of comorbidities variables is a major limitation. From an analysis point, we recoded the comorbidity variable to include the general comorbidity question and specific disease questions, and concluded comorbidities were reported (as present or absent) in 19,680 cases (21.4%) and were missing in 72,371 (78.6%) of cases. Furthermore, patients who were not missing comorbidity were, like you pointed out, more likely to also not be missing symptoms. In retrospect, this is clearly inadequate and the authors amended their objectives, methodology, analysis and consequent conclusions to minimize these limitations. The missingness analysis showed statistically significant differences between sexes, districts and months in comorbidity reporting and reported comorbidity information decreased as the pandemic progressed. This confirms comorbidities are underreported in our study. In light of the high rate and pattern of missingness observed, the authors abstained from

comparing results obtained from these variables and excluded missing information from the calculated percentages. Information forewarning to the level of missingness has been included in the abstract and also at the beginning of the discussion. - [Comment 5] The high level of missingness should entail very cautious interpretation of associations and causation. Because missingness across multiple variables in a data record is not random, spurious associations can emerge. Pediatric patients who were not missing comorbidity data would also be much more likely to NOT be missing symptom data. Response: The authors would like to thank the reviewer for this comment. Upon reflection over our previous analysis, and as previously recognized by the authors, spurious associations can become apparent and probably have... Thank you for your thorough review and comments on this issue. We updated our study and manuscript to minimize this bias and opted to simply describe these variables as results, avoid comparisons between them and concomitantly describe our limitations. - [Comment 6] Furthermore, the authors should conduct and report a systematic analysis of the pattern of missingness by age and gender, and possibly by geographic area and time period as well. Did the quality of surveillance data reporting improve over time? Was it higher or lower in major cities vs. rural areas? Response: The authors would like thank the reviewer for suggesting a systematic analysis of the pattern of missingness which we have included in our manuscript. This analysis has portrayed very interesting results, showing statistical differences between month and district for disease presentation and between month, district and sex for the presence of comorbidities. Two districts presented the highest recorded rates of information: Lisbon, a mainly urban district and Faro. The districts with the lowest recorded information were mainly rural. Furthermore, as the pandemic progressed the missingness rates increased, as discussed in the manuscript. The authors are grateful for this suggestion and hope this analysis can prove an important tool to improve surveillance data in our country. 2. Case counts vs. incidence - [Comment 7] The authors conflate numerator counts of cases with "incidence." Incidence must be expressed as a rate, with an appropriate census-derived population denominator. This is a very elementary error. The authors describe incidence increasing by single years of age in the text, when the data only show increasing numerator counts - there are no incidence rates presented anywhere in the paper. Similarly, the map and discussion of geographic patterns is based on numerators only. Obviously there will always be a larger number of cases in cities than in rural areas. The authors have not investigated whether the incidence RATES are higher in urban vs. rural areas. Response: The authors would like to thank the reviewer for this comment. We have enhanced the analysis to include a census-derived population denominator when characterizing the cases sex and age, we have also calculated hospital and PICU admissions similarly. Census derived population information is, unfortunately, not available to compare with reported case counts geographically. SINAVE reports geographic locations based on district information, which is the portuguese traditional and commonly used location referencing, whereas Portugal Census geographic information is collected as NUTS. NUTS were created by Eurostat with the aim at harmonizing statistical data between european countries for statistical purposes. Nevertheless we have updated the manuscript to include incidence rates, as suggested, with important quality gains in our analysis. 3. Hospitalization and mortality analyses and discussion - [Comment 8] These data are not contextualized at all. How complete was the reporting of pediatric hospitalizations? Were these data directly linked to the case surveillance data, or did they come from an independent, unlinked data source? Is child/family socioeconomic position a barrier to hospitalization in Portugal? Does access to medical care vary geographically? Response: The authors would like to thank the reviewer for these comments. We did not have access to this data or database, we only had access as results published by the National

Health Authority on a document focusing paediatric COVID-19 vaccines, as referenced. So this information is not linked to the epidemiologic surveillance database. In Portugal, free healthcare is accessible to all children under the age of 18 years old. Public SNS hospitals cover all our territory, and despite a growing private sector healthcare, paediatric care is freely available throughout the country. The authors have added hospitalization information to the methods section. - [Comment 9] Hospitalization rates should be presented two ways - population rates (with the child population at risk as the denominator) and case rates (with the child cases as the denominator). Response: The authors thank the reviewer for this comment and we have added information on hospitalization population rates to our manuscript. - [Comment 10] Infants should be separated from the 0-4 group, because both hospitalization and death rates are known to be much higher for infants than for children 1-4 years old. Response: The authors would like to thank the reviewer for this comment and share the reviewer's opinion on the need to separate this age group in two brackets, as we have in our reported case analysis. However, we did not have access to this data, only to published results, so we were unable to separate the infants from the 1-4 years old as recommended. - [Comment 11] The quality of the death data should be described and discussed - what is known about out-of-hospital COVID-19 mortality, misclassification of cause of death (especially in the early days of the pandemic), and any urban-rural differences in quality of mortality surveillance? Response: The authors would like to thank the reviewer for these comments. Portugal has a robust death reporting system, all in-hospital and out-of-hospital deaths are legally required to be certified on the SICO digital platform by a physician. DGS collects death information from all deaths in Portugal in SICO, a national and webbased information system, which includes information on causes of death and related comorbidities, inputted by medical doctors. In the early days of the pandemic much effort was made to avoid the misclassifying causes of death. Over this period, all patients with COVID-19 were referenced to two level 3 hospitals for care, subsequently free testing was available throughout the country, so any ill child could be tested and COVID-19 specific health patient access was implemented. Online health support was available by the SNS24 line and children who presented with fever or other symptoms compatible with COVID-19 were tested. Hospital and PICU admitted patients were regularly tested for COVID-19. 4. Data Tables - [Comment 12] As stated above, many of these tables should be reformulated with incidence rates, not just numerator case counts. Response: The authors thank the reviewer for this comment. We added incidence rates to the general characteristics table. - [Comment 13] In Table 1, cases who were missing gender need to be accounted for in the table. Response: The authors thank the reviewer for this comment. The missing gender data has been added to the table, as a footnote. - [Comment 14] In Table 2, the total number of cases in each age group should be included in the top header row of the table. Response: The authors thank the reviewer for this comment. The authors have added the total number of cases in each age group in the top header row of the table. - [Comment 15] In Table 3, the percents reported in the Female and Male columns are "row percents" and they should be "column percents" to match the way the percents are calculated in the Total column. Also include the totals for females and males in the header row. Response: The authors thank the reviewer for this comment. The authors have corrected the percentage to column percents and have added the total number of females and males in the header row. - [Comment 16] Supplemental Table 2 is very misleading. The majority of cases with missing symptom reports are not included here. The conservative approach would be to assume those cases were asymptomatic - at the very least they should be included as a separate column. The percents as currently shown are uninformative. They should be replaced with row percents, to answer the question "What percent of cases at this year of age had symptoms

reported in the data system?" Response: The authors thank the reviewer for this comment. We've removed tables where crosstabulation between patient presentation and comorbidities was present. This was performed to minimize our exposure to the missingness of the variables involved. Concomitantly we have reviewed table 2 in accordance with your suggestion

REVIEWER 3 GENERAL - [Comment 1] This is an important documentation. However, no information was available for a large number of patients (>62%). These patients have also been included in denominator for calculating the percentage of patients having various symptoms. This is not appropriate. Some recalculations are needed. Response: The authors would like to thank the reviewer for recognizing the importance of this study and for these comments. Our main goal was to describe pediatric COVID-19 presentations and outcomes of patients in our country. Unfortunately, the surveillance data utilized in this analysis had high levels of missingness which greatly hampered part of the analysis conducted. We have updated our analyses in keeping with your feedback. The calculation of the percentages of cases has excluded missing results and furthermore a missingness analysis was conducted. - [Comment 2] More information about patients with MIS-C and death would be better. Response: We thank the reviewer for this comment and despite our efforts and willingness to add information regarding these patients, we are unfortunately limited by the data and information available. TABLES - [Comment 3] In Table-2, provide total number of patients studied in each age group. Response: The authors thank the reviewer for this comment. The authors have added the total number of cases in each age group in the top header row of the table. - [Comment 4] In Table-3, the calculations of percentages are confusing. For example, the number of co-morbidities under the column Total are 2% but it becomes 42.6% for Females and 57% for Males. Response: The authors thank the reviewer for this comment. The authors have corrected the percentage to column percents and have added the total number of females and males in the header row

REVIEWER	Reviewer name: Dr. Mariana Poppe Institution and Country: Hospital Beatriz Angelo, Portugal Competing interests: None
REVIEW RETURNED	20-Jun-2022
GENERAL COMMENTS	GENERAL COMMENT The authors took the previous reviews into consideration, leading to a change in the focus and objective of the article, with an overall improvement in all sections. Small revisions are needed. ABSTRACT The changes made to the abstract are appropriate. In the conclusions section, I would be cautious about the affirmations on clinical presentation, since, as portrayed in the results, missing data makes it difficult to conclude anything about it. I would advise against mentioning it in the conclusions. BACKGROUND Page 7, line 30: "A small number of children needed hospital admission, and of these, a minority required ICU admission either due to severe COVID-19 or multisystem inflammatory disease in children (MIS-C)." This added sentence should have a reference to support it.

MATERIAL AND METHODS

Page 8, line 43: "The National Health Service Hospitals (SNS), freely cover the paediatric population in the country." I'm not sure I understand the meaning of this sentence in this context. Does this mean admissions in hospitals other than the ones from the NHS were excluded? The previous sentence mentions "admissions to all hospitals in Portugal". I would recommend reviewing these phrases to avoid confusion.

Variables - much clearer than the first manuscript, well done.

RESULTS

Page 10, line 5: "An overall reported case incidence of 5.4 per 100 children and adolescents, 5.3% males and 5.5% females." I would recommend changing to 'The overall reported case incidence was 5.4 ...'

Page 10, line 51: "From cases with information on clinical presentation, 13,077 (27.5%) were asymptomatic and 34,393 (72.5%) symptomatic. Specific symptoms were reported in 21,077 (44.4%) of cases and a further 13,316 (14.5%) were recorded as symptomatic without reporting symptoms." Reading this paragraph, the percentages 44.4% and 14.5% seem to not make sense - reading back, one can calculate that they refer to the whole population of symptomatic and asymptomatic cases. In my opinion, the authors could refer to it only among the symptomatic presentations (34,393), as they are characterizing this population only, not the asymptomatic ones (this would mean that 61.3% presented with specific symptoms and 38.7% with symptoms but without further report).

Page 11, line 5: "exclusively included" - I would recommend the alteration to 'included only' to avoid confusion.

Page 11, line 34: "Information on the presence or absence of comorbidities was reported in of cases and there was no information regarding the presence or absence of symptoms in 72,371 (78.6%)." Do the authors mean absence of comorbidities instead of symptoms?

Page 11, line 38: "The three most frequent comorbidities associated with COVID-19 in children and adolescents". The comorbidities aren't necessarily associated with COVID-19, they are the most frequent ones reported in children with COVID-19.

Page 11, line 55: "Disease presentation presented statistical significance between districts and months; whereas comorbidities exhibited significant differences between sexes, district and months." It should be made clear that the authors mean statistical significance in the missing data about the disease presentation and not a difference in the actual disease presentation. Also: is the difference between sexes relevant? In table 4 the percentages are 78.7% and 78.6% - which test was used to calculate this statistical significance? What do the authors take from this supposed difference?

DISCUSSION

Page 15, line 45: "These results probably reflect a higher testing rate of asymptomatic or mildly ill children in Portugal." What is the reference to support this statement? If it is the opinion of the authors, it should state 'might reflect' instead of 'probably reflect', since no evidence was presented to support it.

Page 15, line 51: "Four cases (females under the age of one year

	and presenting comorbidities; 12.5%)". Formulation is unclear. Suggestion: 'Four cases (12.5%), all females under the age of one year with comorbidities, ...' CONCLUSIONS Page 17, line 19: "The clinical manifestations of disease reported in our analysis were predominantly respiratory and neurologic, while gastrointestinal symptoms were infrequent." I wouldn't recommend mentioning this in the conclusions, since it is no longer the focus of the article and is hindered by missing data, as the authors admit. TABLE 4 – The authors should mention the statistical test used in this analysis (footnotes).
--	---

REVIEWER	Reviewer name: Dr. Harish PEMDE Institution and Country: Kalawati Saran Children's Hospital, India Competing interests: None
REVIEW RETURNED	10-Jun-2022
GENERAL COMMENTS	The article has improved a lot. This calls for looking into the process of data collection for public health goals and improving the systems for it.

VERSION 2 – AUTHOR RESPONSE

Reviewer 1 1. The article has improved a lot. This calls for looking into the process of data collection for public health goals and improving the systems for it. The authors would like to thank the reviewer for this comment. We feel the peer review process has been extremely positive to the overall quality of our analysis. This study has highlighted the need to improve surveillance data in Portugal and measures need to be put in place in order for this system to be improved. Simultaneously we described, as best as we could, and given the data limitations, paediatric COVID-19 cases in children and adolescents in our country

Reviewer 2 1. The authors took the previous reviews into consideration, leading to a change in the focus and objective of the article, with an overall improvement in all sections. Small revisions are needed. The authors would like to thank the reviewer for this comment. We feel the peer review process has been extremely positive to the overall quality of our analysis. The authors wanted to describe pediatric COVID-19 cases and outcomes however given the data limitation found we did shift our focus toward improvement of information systems towards public health goals. This study has highlighted the need to improve surveillance data in Portugal and measures need to be put in place in order for this system's improvement. **ABSTRACT 2.** The changes made to the abstract are appropriate. In the conclusions section, I would be cautious about the affirmations on clinical presentation, since, as portrayed in the results, missing data makes it difficult to conclude anything about it. I would advise against mentioning it in the conclusions. The authors would like to thank the

reviewer for this comment. We have updated our manuscript accordingly; symptomatic descriptions have been excluded from the conclusion's sections. BACKGROUND 3. Page 7, line 30: "A small number of children needed hospital admission, and of these, a minority required ICU admission either due to severe COVID-19 or multisystem inflammatory disease in children (MIS-C)." This added sentence should have a reference to support it. The authors would like to thank the reviewer for this comment. We have updated our manuscript accordingly and have added the needed referencing. MATERIAL AND METHODS 4. Page 8, line 43: "The National Health Service Hospitals (SNS), freely cover the paediatric population in the country." I'm not sure I understand the meaning of this sentence in this context. Does this mean admissions in hospitals other than the ones from the NHS were excluded? The previous sentence mentions "admissions to all hospitals in Portugal". I would recommend reviewing these phrases to avoid confusion. The authors would like to thank the reviewer for this comment. This sentence is meant to provide information regarding the Portuguese Health system as a freely accessible system where there aren't significant financial barriers to access and opposing other countries where health coverage is not universal and free for children. We have referenced the total national hospital admission numbers from the DGS report and these are referred to total hospital admissions. 5. Variables - much clearer than the first manuscript, well done. The authors would like to thank the reviewer for these comments. RESULTS 6. Page 10, line 5: "An overall reported case incidence of 5.4 per 100 children and adolescents, 5.3% males and 5.5% females." I would recommend changing to 'The overall reported case incidence was 5.4 ...' The authors would like to thank the reviewer for this comment. We have updated our manuscript accordingly. 7. Page 10, line 51: "From cases with information on clinical presentation, 13,077 (27.5%) were asymptomatic and 34,393 (72.5%) symptomatic. Specific symptoms were reported in 21,077 (44.4%) of cases and a further 13,316 (14.5%) where recorded as symptomatic without reporting symptoms." Reading this paragraph, the percentages 44.4% and 14.5% seem to not make sense - reading back, one can calculate that they refer to the whole population of symptomatic and asymptomatic cases. In my opinion, the authors could refer to it only among the symptomatic presentations (34,393), as they are characterizing this population only, not the asymptomatic ones (this would mean that 61.3% presented with specific symptoms and 38.7% with symptoms but without further report). The authors would like to thank the reviewer for this comment. We have updated our manuscript accordingly; it does make sense for the denominator to be the symptomatic presentations, either with and without characterization of specific symptoms. 8. Page 11, line 5: "exclusively included" - I would recommend the alteration to 'included only' to avoid confusion. The authors would like to thank the reviewer for this comment. We have updated our manuscript accordingly. 9. Page 11, line 34: "Information on the presence or absence of comorbidities was reported in of cases and there was no information regarding the presence or absence of symptoms in 72,371 (78.6%)." Do the authors mean absence of comorbidities instead of symptoms? The authors would like to thank the reviewer for this comment. We have corrected our manuscript accordingly. 10. Page 11, line 38: "The three most frequent comorbidities associated with COVID-19 in children and adolescents". The comorbidities aren't necessarily associated with COVID-19, they are the most frequent ones reported in children with COVID-19. The authors would like to thank the reviewer for this comment. We have updated our manuscript accordingly. 11. Page 11, line 55: "Disease presentation presented statistical significance between districts and months; whereas comorbidities exhibited significant differences between sexes, district and months." It should be made clear that the authors mean statistical significance in the missing data about the disease presentation and not a difference in the actual disease presentation. Also: is the difference between

sexes relevant? In table 4 the percentages are 78.7% and 78.6% - which test was used to calculate this statistical significance? What do the authors take from this supposed difference? The authors would like to thank the reviewer for this comment. This section has been removed from our analysis as suggested by the editor. The authors used the ChiSquare test to compare these groups on the missingness analyses. Despite the small difference in reporting rates, it is statistically significant. It would be very interesting to further analyze if at the pediatric age there are systematic statistically significant differences in reporting comorbidities according to sex on health surveillance data and in other data sources. As we know gender disparities have been reported in health practice overall. Given this being an isolated result no conclusions can be established, however the authors had highlighted this and are planning to review other databases. DISCUSSION 12. Page 15, line 45: "These results probably reflect a higher testing rate of asymptomatic or mildly ill children in Portugal." What is the reference to support this statement? If it is the opinion of the authors, it should state 'might reflect' instead of 'probably reflect', since no evidence was presented to support it. The authors would like to thank the reviewer for this comment. We have updated our manuscript accordingly. 13. Page 15, line 51: "Four cases (females under the age of one year and presenting comorbidities; 12.5%)". Formulation is unclear. Suggestion: 'Four cases (12.5%), all females under the age of one year with comorbidities, ...' The authors would like to thank the reviewer for this comment. We have updated our manuscript accordingly. CONCLUSIONS 14. Page 17, line 19: "The clinical manifestations of disease reported in our analysis were predominantly respiratory and neurologic, while gastrointestinal symptoms were infrequent." I wouldn't recommend mentioning this in the conclusions, since it is no longer the focus of the article and is hindered by missing data, as the authors admit. The authors would like to thank the reviewer for this comment. We have updated our manuscript accordingly. 15. TABLE 4 – The authors should mention the statistical test used in this analysis (footnotes). The authors would like to thank the reviewer for this comment. This table has been removed for our manuscript and the results were argued in the discussion section